# Pain drawing as a screening tool for anxiety, depression and reduced health-related quality of life in back pain patients: A cohort study

Katharina Weßollek[1,2]*, Ana Kowark[1], Michael Czaplik[1], Rolf Rossaint[1], Pascal Kowark[1]

1 Department of Anaesthesiology, Medical Faculty, University Hospital RWTH Aachen, Aachen, Germany,
2 Department of Dermatology and Allergology, Medical Faculty, University Hospital RWTH Aachen, Aachen, Germany

* kwessollek@ukaachen.de

## Abstract

### Background

Back pain patients are more likely to suffer from depression, anxiety and reduced quality of life. Pain drawing is a simple, frequently used anamnesis tool that facilitates communication between physicians and patients. This study analysed pain drawings to examine whether pain drawing is suitable as a screening tool for signs of anxiety, depression or reduced quality of life, as the detection of these symptoms is essential for successful treatment.

### Methods

Pain drawings of 219 patients with lower back pain were evaluated retrospectively. Pain drawings are a schematic drawing of a human being. Six variables of the pain drawing were analysed. Subscales of the Hospital Anxiety and Depression Scale (HADS) and the Mental Component Summary (MCS) of the Short Form 12 (SF-12) were used to measure anxiety, depression and quality of life, respectively. Descriptive statistics, uni- and multivariate linear regression analyses and analysis of variance were performed. Logistic regression analyses were conducted for suitable variables.

### Results

We revealed significant positive correlations between the variables "filled body surface" and "number of pain sites" and the anxiety (HADS-A) and depression subscales (HADS-D) of the HADS (p<0.01). The same predictors had significant negative correlations with the MCS (p<0.01). However, the sensitivity and specificity of the variable "number of pain sites" were too low compared to those for existing screening tests to consider it as a screening tool for anxiety, depression and quality of life (HADS-A: sensitivity: 45.2%, specificity: 83.3%; HADS-D: sensitivity: 61.1%, specificity: 51%; MCS: sensitivity: 21.2%, specificity: 85.7%).

**Data Availability Statement:** Data may be shared upon request after signing a data transfer agreement. According to our data protection law we are not allowed to publish data containing

sensitive patient information. Data requests may be sent to the Chief Information Security Officer (CISO) of the RWTH University Hospital Aachen, Germany. Contact: ISB@ukaachen.de.

**Funding:** The author(s) received no specific funding for this work.

**Competing interests:** The authors have declared that no competing interests exist.

## Conclusions

There were significant correlations between the amount of filled body surface and the number of pain sites in the pain drawing and anxiety, depression and quality of life. Although useful in routine clinical practice, pain drawing cannot be used as a screening tool based on our results.

## Introduction

Pain drawing is an anamnesis tool of minimal complexity. The first documented pain drawing dates back to the first half of the 16th century [1]. Based on the work published by Palmer in 1949, pain drawing became part of modern clinical routine at that time [2]. Today's pain drawings usually consist of a schematic drawing of the outline of a human figure from ventral and dorsal perspectives. The patients are asked to sketch in their pain. Further notes might be added in a free text box.

Pain drawing is well accepted by patients; the instructions are easy to understand and the task can be completed quickly. Unlike questionnaires, pain drawing is accessible regardless of individual language competence and other personal characteristics.

Pain drawings are important because they increase physicians' understanding of patients' pain [3]. One study showed that patients favoured pain drawing over a written description of their pain in terms of correct interpretation by the physician [4].

Since pain drawing is widely used in routine clinical practice, it would be helpful if this simple anamnesis tool could also be used to examine other symptoms frequently associated with back pain.

Patients with lower and chronic back pain are more likely to suffer from depression, anxiety and reduced health-related quality of life than the normal population, which may impair their recovery [5–9].

Back pain poses a growing challenge for German health care facilities. Studies show that approximately 4% of the workforce is lost every year due to absenteeism from work as a result of back pain [10].

Evidence suggests that counselling might significantly reduce sick leave in lower back pain patients [11]. Early identification of patients who are suffering from a psychiatric-psychological disorder or reduced quality of life and timely referral to specialists for treatment are essential. This early identification might induce a significant reduction in psychological strain and the duration of suffering [12, 13].

To date, studies examining the usability of pain drawings as screening tools for psychiatric-psychological comorbidities have shown inconsistent results. Some studies on patients with lower back and musculoskeletal pain have reported a statistically significant correlation between pain drawing variables, such as pain extent and pain site, and psychological characteristics, such as depression and somatisation [14–16]. Contrary, other studies mainly focussing on lower back pain patients have not found significant results [17–19]. A systematic review and meta-analysis published in 2015 concluded that the use of qualitatively analysed pain drawings as a screening tool for psychiatric conditions in lower back pain patients currently cannot be recommended without restriction [20]. Nevertheless, the authors acknowledged that future meta-analyses might come to a different result by using higher quality and further studies [20].

In this study new pain drawing variables, namely, "hatching degree" and "total word count (TWC)" were added to previously investigated pain drawing variables. We hypothesised that a

combination of pain drawing variables can be used as a screening tool for signs of reduced health-related quality of life, depression or anxiety in patients with lower back pain.

## Materials and methods

This was a retrospective cohort study performed at University Hospital RWTH Aachen, Germany. This study was reported in accordance with the STROBE statement (S1 Table) [21]. The sample size was calculated to detect medium to large effects [22]. The Ethics Committee at the Medical Faculty of the RWTH University of Aachen, Germany, approved a waiver of consent for this retrospective study (EK 187/18) in July 2018. Data were pseudonymised and assessed from July to November 2018. The study analysed the records of 219 patients. The inclusion criteria were the diagnosis of lower back pain and an initial appointment at the pain medicine outpatient department of the institution between 2013 and 2018. Exclusion criteria comprised legally incompetent patients and patients with sensory limitations, such as blindness. We aimed to control for selection bias by consecutive inclusion of all eligible patients into this secondary analysis.

Pain drawing and all outcomes are part of the Deutscher Schmerzfragebogen (DSF, German pain questionnaire) [23]. Usually, patients receive DSF by mail before the first appointment and complete it at home.

The main focus of this study was to analyse whether drawing pain facilitates the early detection of signs of anxiety, depression and reduced quality of life.

Pain drawing is a schematic drawing of the outline of a human figure from ventral and dorsal perspectives. In addition, a free text box offers space for further pain-related notes. The accompanying instructions are short and easily understood ("barrier-free"). Patients are asked to sketch in their pain.

This study examined six variables derived from pain drawing. The variable selection was based on a comprehensive review of previous literature and the authors' clinical experience with pain drawings. The authors introduced two new variables, "hatching degree" and "total word count (TWC)", for pain drawing analysis.

To ensure comparability, all pain drawings were converted into binary black and white images.

In the following, the variables examined are explained in detail:

- Variable 1 "Filled body surface":

This variable evaluates the proportion of the human figure outline marked by the patient. The measurement was performed with computer-aided ImageJ software (Wayne Rasband, National Institute of Health, Bethesda, Maryland, USA). The ratio of filled body surface to total body surface was calculated; therefore, the unit of measurement was percent. Markings beyond the outline of the human figure were not taken into account (Fig 1).

- Variable 2 "Hatching degree":

Hatching degree represents the intensity of a marking. The average degree of hatching was calculated (Fig 1). The possible value range specified by the ImageJ software was from 0 (black) to 255 (white).

- Variable 3 "Number of pain sites":

The pain drawing was divided into seven regions (right leg, left leg, right arm, left arm, thorax/abdomen, back and head). Each region marked in any way was classified as a pain site (Fig 1).

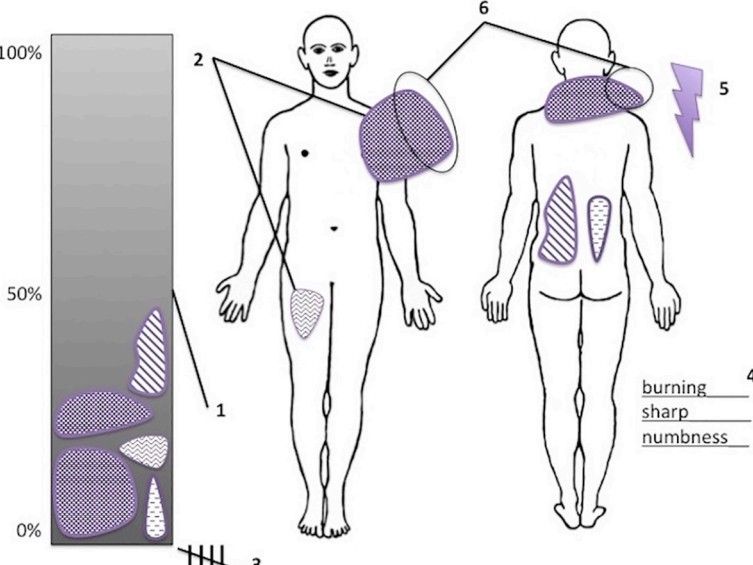

**Fig 1. Fictitious example of a pain drawing.** Modified pain drawing based on DSF. For better illustration, this fictitious pain drawing is presented in colour. Pain drawings in the study were binarily transformed into black and white images before analysis. Variable 1 "Filled body surface": The total area of the human figure is 100%. The proportions of the patients' markings within the outline were assessed as percentages. In the diagram on the left this process is visualised not true to scale. Variable 2 "Hatching degree": Presented are a lighter hatching (right upper thigh, ventral) and a darker hatching (left shoulder, ventral). Variable 3 "Number of pain sites": This variable is calculated by adding up the number of pain sites. In this example, five pain sites are presented (right leg, thorax/abdomen, left arm, head and back). Variable 4 "Total word count": In this pain drawing, there are three words describing the pain. Variable 5 "Use of indicative symbols": In the dorsal view of the human figure, there is a lightning bolt drawn next to the right shoulder as an indicative symbol. Variable 6 "Crossing the outlines": In the two areas marked with the number 6, the outline of the human figure is crossed.

- Variable 4 "Total Word Count (TWC)": This variable assessed the number of words in the free text box (Fig 1).

- Variable 5 "Use of indicative symbols":

  This variable was dichotomous. Any use of an indicative symbol (e.g. arrow or lightning bolt) met the condition (Fig 1).

- Variable 6 "Crossing the outlines":

  This variable was dichotomous. Any crossing of the outlines of the human figure met the condition (Fig 1).

  The selected variables were used in the regression model as independent variables (predictors or predictor variables).

  The primary aim of this study was to investigate whether drawing pain is a proper tool to quickly screen for signs of depression, anxiety or reduced quality of life in patients with lower back pain. To examine this hypothesis, we used the scores of various validated screening questionnaires investigating the aforementioned diseases as reference values; to identify the presence of depression and anxiety, the Hospital Anxiety and Depression Scale (HADS) was used [24]. The HADS has two independently scored scales quantifying symptoms of anxiety and depression, with the HADS-A and HADS-D, respectively. Both scales were used in this study, since the patients had completed them in the routine clinical examination at baseline. Each subscale has 7 items and each item contains a statement. Patients had to rate their agreement

with the statement on a four-point Likert scale (0–3). Scores range from 0 to 21 points for each subscale [25]. The cut-off value used in this study for each HADS subscale was $\geq 11$ points, according to the DSF manual. Scores $\geq 11$ points indicate the presence of a mood disorder [25].

The degree of impairment of health-related quality of life was identified by examining the scores on the Short Form 12 (SF-12) [26]. The SF-12 consists of 12 items. The items include statements and questions about physical and mental health. The SF-12 has a physical and a mental component summary, which are scored independently; this study solely investigated the mental component summary (MCS). The cut-off value used in this study for the MCS was $< 44$ points. According to the DSF manual, scores $< 44$ points indicate a low health-related quality of life.

## Statistical analysis

All statistical analyses were performed using Microsoft Excel 2010 (Microsoft Corporation, Redmond, Washington, USA) and IBM SPSS 25 (IBM Corporation, Armonk, New York, USA). A descriptive data evaluation of the patient cohort was performed.

Absolute and relative frequencies were calculated for the categorical variables sex, the degree of chronification, according to von Korff [27], pain progression, pain duration, medication and the presence of psychological comorbidity. The continuous descriptive variables age, the number of involved medical disciplines thus far and the number of previous therapy measures were examined with regard to the mean value and standard deviation (SD).

For the outcomes of HADS-A, HADS-D and MCS, the number of available datasets of the patient collective was determined. For all outcomes, the mean value, SD and confidence intervals (CIs) were determined.

The potentially confounding variables age, sex and pain duration were assessed with correlation and point biserial correlation, independent T-test and chi-square test. Partial correlations were obtained.

Univariate and multivariate linear regression analyses were performed. If model assumptions were not adequately fulfilled, the following procedures were used: bootstrapping, which is used in linear regression to determine CIs and p-values, which are not dependent on the normal distribution of residuals and/or homoscedasticity. Furthermore, if the homoscedasticity assumption was violated, the weighted least squares (WLS) method was used to calculate robust regression coefficients. Influence and outlier diagnostics were performed. Linear regression was conducted controlling for potential confounders.

To determine the influence of the categorical (nominal) predictors, analysis of variance (VA, ANOVA) was used.

Finally, logistic regression analyses were performed for the suitable independent variable (the number of pain sites). For logistic regression the outcome variables were grouped according to their cut-off values defined in the DSF manual. Influence and outlier diagnostic were performed.

Missing datasets were considered missing at random. No imputation was performed, and all analyses were based on complete cases [21].

## Results

### Patient characteristics

Data from 219 patients were collected, with 121 (55.3%) being female. The overall mean age was 57.8 (15.9) years, with females and males being at a mean age of 58.8 (16.9) and 56.7 (14.6)

**Table 1. Number of datasets and scores.**

| | | HADS-A | HADS-D | MCS |
|---|---|---|---|---|
| Total | Obtained datasets | 207 | 214 | 167 |
| | Missing datasets | 12 | 5 | 52 |
| | Arithmetic mean | 9.92 | 10.61 | 41.39 |
| | Median | 10.00 | 11.00 | 39.68 |
| | Standard deviation (SD) | 4.74 | 4.68 | 12.69 |
| Females | Arithmetic mean | 10.27 | 10.57 | 40.18 |
| | N | 113 | 118 | 95 |
| | Std. Deviation | 4.51 | 4.36 | 12.65 |
| Males | Arithmetic mean | 9.50 | 10.67 | 42.98 |
| | N | 94 | 96 | 72 |
| | Std. Deviation | 4.98 | 5.07 | 12.66 |

HADS-A: Hospital Anxiety and Depression Scale-Anxiety; HADS-D: Hospital Anxiety and Depression Scale-Depression; MCS: mental component summary of the Short Form 12 questionnaire.

Cut-off value HADS-subscales: $\geq$ 11 points. Cut-off value MCS: < 44 points.

years, respectively. There was no significant difference in mean age between males and females (t(216.03) = -0.95, p = 0.341, S2 Table).

Of the 219 patients, 136 (62.1%) had been suffering from pain for more than two years. Half of the patients (50.9%) were suffering from mental stress.

The average scores and average scores by sex for both the HADS and the MCS are presented in Table 1. The average subscale scores were close to the cut-off values (HADS-A 9.92 points; HADS-D 10.61 points; MCS 41.39 points). Subscale scores did not differ significantly between sexes (HADS-A: $M_{female}$ = 10.27, $M_{male}$ = 9.50, t(189.80) = -1.15, p = 0.252; HADS-D: $M_{female}$ = 10.57, $M_{male}$ = 10.67, t(188.24) = 0.15, p = 0.880; MCS: $M_{female}$ = 40.18, $M_{male}$ = 42.98, t (152.94) = 1.41, p = 0.160; S3 Table). S1 and S2 Figs show mean outcomes in different age categories of both sexes.

Neither age nor sex correlated significantly with predictors or outcomes. We detected significant positive correlations between pain duration and the predictors "filled body surface" and "number of pain sites" as well as the outcomes of HADS-A and HADS-D, respectively. Further, pain duration was associated with the predictor "use of indicative symbols". Therefore, pain duration was considered a potential confounding variable. Partial correlation revealed that controlling for pain duration had only a small effect on correlation coefficients and did not erase significance.

## Predictive pain drawing variables for psychological outcomes

Table 2 presents the univariate linear regression analyses to determine predictive pain drawing variables for psychological outcomes. Significantly associated with all three psychological variables, HADS-A, HADS-D and MCS, were the variables "filled body surface" and "number of pain sites", respectively (Table 2). The association was positive for HADS-A and HADS-D, whereas it was negative for MCS.

In multivariate regression analyses, a significant association was found between the predictor "number of pain sites" and the HADS-A score (p = 0.001), and the predictor "filled body surface" and the HADS-D score (p = 0.005) (Table 3). The other associations lost their significance (Table 3). Effects were stable when controlling for pain duration. However, there was a non-significant negative relationship between the predictor "number of pain sites" and the MCS score (p = 0.051).

**Table 2. Linear regression models for the dependent variables HADS-A, HADS-D and MCS.**

| Dependent variable | Independent variable | B | Std. Error | Sig. (2-tailed) | 95% CI |
|---|---|---|---|---|---|
| HADS-A | Filled body surface | 0.23 | 0.05 | 0.001* | 0.14 to 0.32 [a] |
| | Number of pain sites | 0.92 | 0.20 | 0.000* | 0.53 to 1.32 |
| HADS-D | Filled body surface | 0.22 | 0.05 | 0.001* | 0.12 to 0.31 [a] |
| | Number of pain sites | 0.55 | 0.20 | 0.005* | 0.17 to 0.94 |
| MCS | Filled body surface | -0.50 | 0.12 | 0.001* | -0.71 to -0.21 [a] |
| | Number of pain sites | -1.71 | 0.60 | 0.005* | -2.88 to -0.53 |

HADS-A: Hospital Anxiety and Depression Scale-Anxiety; HADS-D: Hospital Anxiety and Depression Scale-Depression; MCS: mental component summary of the Short Form 12 questionnaire.

a. Bias-corrected and accelerated CI; unless otherwise stated, bootstrap results are based on 1000 bootstrap samples.

* $p < 0.05$.

## Logistic regression analysis

For all dependent variables, the independent variables "filled body surface" and "number of pain sites" were significant predictors.

The independent variable "number of pain sites" had at least one data set for all possible combinations of variables. Crosstabulation revealed that 21.4% of variable combinations for the outcomes of HADS-A and MCS had a frequency of less than 5. For the HADS-D outcome, the proportion of variable combinations with a frequency of less than 5 was 14.3%.

**Table 3. Bootstrapping procedure to obtain robust confidence intervals for the multivariate regression models for the dependent variables HADS-A, HADS-D and MCS.**

| Dependent variable | Independent variable | B | Std. Error | Sig. (2-tailed)* | BCa 95% CI [a] |
|---|---|---|---|---|---|
| HADS-A | Filled body surface | 0.09 | 0.07 | 0.180 | -0.05 to 0.20 |
| | Hatching degree | 0.01 | 0.01 | 0.159 | -0.01 to 0.03 |
| | Number of pain sites | 0.92 | 0.24 | 0.001* | 0.46 to 1.37 |
| | Total Word Count | -0.04 | 0.03 | 0.103 | -0.10 to -0.01 |
| | Use of indicative symbols | 0.55 | 1.04 | 0.572 | -1.65 to 2.73 |
| | Crossing the outlines | -0.88 | 0.72 | 0.214 | -2.28 to 0.68 |
| HADS-D | Filled body surface | 0.20 | 0.06 | 0.005* | 0.09 to 0.32 |
| | Hatching degree | 0.001 | 0.01 | 0.921 | -0.02 to 0.02 |
| | Number of pain sites | 0.38 | 0.25 | 0.123 | -0.17 to 0.94 |
| | Total Word Count | -0.03 | 0.02 | 0.101 | -0.08 to -0.01 |
| | Use of indicative symbols | -0.05 | 1.10 | 0.960 | -2.49 to 2.05 |
| | Crossing the outlines | -1.18 | 0.67 | 0.081 | -2.53 to 0.19 |
| MCS | Filled body surface | -0.26 | 0.18 | 0.131 | -0.61 to 0.21 |
| | Hatching degree | -0.03 | 0.03 | 0.380 | -0.08 to 0.02 |
| | Number of pain sites | -1.38 | 0.70 | 0.051 | -2.68 to -0.19 |
| | Total Word Count | 0.05 | 0.08 | 0.442 | -0.04 to 0.26 |
| | Use of indicative symbols | -3.08 | 3.75 | 0.390 | -11.16 to 5.49 |
| | Crossing the outlines | 1.99 | 2.17 | 0.356 | -2.28 to 5.78 |

HADS-A: Hospital Anxiety and Depression Scale-Anxiety; HADS-D: Hospital Anxiety and Depression Scale-Depression; MCS: mental component summary of the Short Form 12 questionnaire.

a. Unless otherwise stated, bootstrap results are based on 1000 bootstrap samples.

* $p < 0.05$.

**Table 4. Logistic regression.**

| Dependent variable | | B | Std. Error | Sig. (p-value) | OR (95% CI) |
|---|---|---|---|---|---|
| HADS-A (categorial) | Number of pain sites | 0.44 | 0.10 | 0.000* | 1.54 (1.27 to 1.89) |
| HADS-D (categorial) | Number of pain sites | 0.23 | 0.09 | 0.011* | 1.26 (1.05 to 1.50) |
| MCS (categorial) | Number of pain sites | -0.22 | 0.11 | 0.036* | 0.80 (0.65 to 0.99) |

HADS-A: Hospital Anxiety and Depression Scale-Anxiety; HADS-D: Hospital Anxiety and Depression Scale-Depression; MCS: mental component summary of the Short Form 12 questionnaire.

*p<0.05.

The logistic regression models represented the data well for all three outcomes (HADS-A: $\chi^2 = 20.98$, p<0.000; $\chi^2$HL = 8.70, p = 0.069; HADS-D: $\chi^2 = 6.79$, p = 0.009; $\chi^2$HL = 0.56, p = 0.968 and MCS: $\chi^2 = 4.70$, p = 0.030; $\chi^2$HL = 2.98, p = 0.561).

**HADS-A.** The logistic regression model explained 13% of the variance (Nagelkerke $R^2$ = 0.13). The variable "number of pain sites" made a significant contribution to the prediction of the outcome (Table 4). The model classified 65.7% of cases into the correct HADS-A category on average. The detection of normal HADS-A scores was a strength of the model (specificity: 83.3%). Most patients with abnormal HADS-A scores were incorrectly classified as normal (sensitivity: 45.2%).

**HADS-D.** The logistic regression model accounted for 4% of the variance (Nagelkerke $R^2$ = 0.04). The variable "number of pain sites" also significantly contributed to the prediction of the correct HADS-D category (Table 4). The sensitivity was 61.1%, the specificity was 51%.

**MCS.** There was 4% of variance explained by the model (Nagelkerke $R^2$ = 0.04). Further, the variable "number of pain sites" significantly contributed to the prediction of the MCS result (Table 4). Averagely 59.8% of cases were classified correctly. The specificity was 85.7%, the sensitivity 21.2%.

## Discussion

In the present study, we analysed pain drawing using a novel combination of known methods to combine quantitative ("filled body surface", "degree of hatching", "number of pain sites" and "total word count") and qualitative ("use of indicative symbols" and "crossing the outlines") variables of pain drawing. Our objective was to evaluate the usefulness of pain drawing as a screening tool for reduced health-related quality of life, depression or anxiety in patients with lower back pain.

While there were some promising significant correlations, especially between the variables "filled body surface" and "number of pain sites" and the outcomes, combining multiple predictors did not improve the explanation of variance.

Sensitivity and specificity were calculated for the predictor number of pain sites. We revealed sensitivities of 45.2% and 61.1% for HADS-A and HADS-D, respectively. The sensitivity of MCS was 21.2%. Thus, we conclude that the number of pain sites cannot be used as a screening tool for the outcomes. The sensitivity and specificity of the HADS are higher and therefore more efficient than the obtained scores [28, 29]. Screening tests usually aim for high sensitivity to identify potential cases. The obtained sensitivity for MCS was 21.2%, which is too low for a screening tool. Moreover, other tests measuring quality of life have better sensitivity and specificity [30, 31].

A meta-analysis published in 2015 attempted to perform additional sensitivity analyses for studies that found a correlation between pain drawings (analysed with qualitative evaluation methods) and psychological abnormalities in lower back pain patients [20]. This meta-analysis

comprised seven studies. The quality of the studies was rated as poor to moderate. The average sensitivity of the qualitative evaluation methods was 45% (range: 4–86%), and the average specificity was 66% (range: 40–100%).

A similar systematic review from 2006 revealed that approximately half of the 19 studies examined found a correlation between pain drawing and psychological status [17]. The included studies analysed quantitative and qualitative variables of pain drawing and anxiety and depression. Nevertheless, only five of the 19 included studies provided data on predictive power. Additionally, the statistical power of most of the included studies was low. The values for sensitivity and specificity subsequently determined by the authors were too heterogeneous for six of the 19 studies to make a clear recommendation for the use of pain drawings as a screening tool. The sensitivity ranged from 24–93% (median: 56%) and the specificity ranged from 48–91% (median: 79.5%) [17].

Apart from satisfactory sensitivity and specificity, a screening tool should be time- and cost-efficient. Assessing a pain drawing takes more time than scoring a questionnaire. This is another reason why pain drawing is not favourable as a screening tool.

The comparability of the results of former studies analysing the association of psychological-psychiatric comorbidities and pain drawing is strongly limited by the heterogeneity of the selected outcomes. The aforementioned systematic review stated that 20 different test procedures were used to assess the psychological state in the 19 examined studies [17]. One way to reduce the lack of comparability resulting from heterogeneous outcome values is to choose test instruments that are already known and frequently used. Therefore, we used the commonly applied test instruments HADS and SF-12.

Not only the heterogeneity of the outcome made it difficult to compare studies, but also the different methods for evaluating pain drawings. The methods to analyse pain drawings known thus far are subject to three basic principles.

The first principle evaluates the presence of various qualitative characteristics that have been defined by the authors as typical for abnormal pain drawings. The presence of these characteristics is assessed by a scoring system. One of the earliest approaches to evaluating pain drawings, the so-called Penalty Point System (PPS), according to Ransford, is an example of a method following this principle [32].

The second principle is quantitative and measures filled body surfaces. The weighted body surface system (BS) method, according to Margolis, originally from burn surgery, follows this principle [33].

The third principle is the analysis of the number of pain sites. An example of this principle is "Pain Sites Scoring" [34].

To our knowledge, no other study has attempted to combine qualitative and quantitative pain drawing scoring systems to screen for anxiety, depression and quality of life in lower back pain patients. However, a study conducted on whiplash patients compared the modified PPS and a method of body surface assessment. The body surface method made a better contribution to the regression model than the PPS [35]. Our study revealed that the quantitative variables were also more suitable than the qualitative variables.

The patients in this study had a high prevalence of anxiety and depression, which is in line with previous research [36]. In particular, patients with chronic back pain have a significantly increased psychological burden and a reduced quality of life compared to the general population [5, 37]. Since the effect has been observed across different populations, we claim that the results of our back pain patient cohort are generalisable to other back pain patient cohorts.

Our data showed no difference between sexes in terms of mean outcomes. Other studies investigating the relationship between sex and depression and anxiety in back pain patients revealed that females suffer from more severe depression, anxiety and stress than males [38,

39]. There was no correlation between outcomes and age, which is in line with a previous study on chronic back pain [40], in which they did not find significant differences in depressed mood among age categories.

Finally, we would like to discuss the statistical analysis. We used crosstabulation before fitting of the logistic regression model. Only for the predictor "number of pain sites" and the outcomes did all columns have a frequency of at least one. The proportion of columns with a frequency < 5 was 14.3% for the variable "number of pain sites" and HADS-D. The proportion of columns with a frequency < 5 was 21.4% for HADS-A and MCS. It is advised to look for proportions < 20% to ensure the goodness of fit [41]. We decided to perform logistic regression since the proportion was close to the advised value. We carefully paid attention to standard errors since an inflated standard error indicates a bad fit of the model [41]. The obtained standard errors were not disproportionally large.

The identification of confounding variables for this study was challenging. A confounder is associated with the outcome and the predictor, leading to potential false conclusions when not taken into account [42]. While it is reasonable to suspect an association between age, sex, and pain duration and the outcomes, there is no reason to assume that these variables are associated with the predictors. We tried to identify potential confounders statistically. There were no correlations between age and sex and the variables of interest; therefore, we did not consider them confounders and did not perform further analyses. There were indicators that pain duration might be a confounding variable, since it correlated with both the outcomes HADS-A and HADS-D and the predictors "filled body surface", "number of pain sites" and "use of indicative symbols". Therefore, we recalculated linear regression analyses and controlled for pain duration. The significant effects remained.

According to the literature, the addition of confounders in logistic regression leads to a precision loss [43]. Therefore, we did not perform analyses to test confounding for the variable "use of indicative symbols".

## Limitations

Notably, we did not analyse the proportions of markings outside of the pain drawing to ensure comparability between pain drawings. Furthermore, we did not perform inter-rater or intra-rater reliability tests. The main limitation of this study is the use of retrospective data. Retrospective data analyses are scientifically weaker than other study designs such as randomised-controlled trials and pose the risk of information bias. Further, this study was not blinded, resulting in a high risk of detection bias. Also, we acknowledge that our results refer to German back pain patients, and large international multicentre studies are warranted.

## Conclusion

This study attempted to develop a new qualitative-quantitative pain drawing analysis method for screening anxiety, depression and health-related quality of life in patients with lower back pain. The novel variables "degree of hatching" and "total word count" did not show any significant correlation with the outcomes, assessed by the HADS-A, HADS-D and MCS. Additionally, merging predictor variables did not lead to an improvement in predictive power.

The present study revealed significant correlations between the variables "filled body surface" and "number of pain sites" and psychological status (anxiety and depression) and health-related quality of life. Sensitivity analysis could be performed only for the variable "number of pain sites"; the obtained results for sensitivity and specificity were quite low compared to existing screening tests. Pain drawing is a useful tool in routine clinical practice. Nevertheless,

based on our results, pain drawing should not be used as a screening tool for signs of anxiety and depression and reduced health-related quality of life in lower back pain patients.

## Supporting information

**S1 Table. STROBE checklist.**
(DOC)

**S2 Table. Independent T-test comparing age means between sexes.** *p<0.05.
(DOCX)

**S3 Table. Independent T-test comparing outcome means between sexes.** *p<0.05.
(DOCX)

**S4 Table. Linear regression models for the independent variables 2 ("hatching degree") and 4 ("total word count") without significant results.** HADS-A: Hospital Anxiety and Depression Scale-Anxiety; HADS-D: Hospital Anxiety and Depression Scale-Depression; MCS: mental component summary of the Short Form 12 questionnaire. a. Bias-corrected and accelerated CI; unless otherwise stated, bootstrap results are based on 1000 bootstrap samples. *p<0.05.
(DOCX)

**S5 Table. Analysis of variance (ANOVA) for the independent variables 5 ("use of indicative symbols") and 6 ("crossing the outlines").** HADS-A: Hospital Anxiety and Depression Scale-Anxiety; HADS-D: Hospital Anxiety and Depression Scale-Depression; MCS: mental component summary of the Short Form 12 questionnaire. *p<0.05.
(DOCX)

**S1 Fig. Mean scores for HADS-A, HADS-D and MCS for different age categories in females.**
(DOCX)

**S2 Fig. Mean scores for HADS-A, HADS-D and MCS for different age categories in males.**
(DOCX)

## Acknowledgments

We would like to thank the Department of Anaesthesiology of the University Hospital RWTH Aachen. We would also like to thank the staff of the central archive of the University Hospital RWTH Aachen. Moreover, special thanks go to the German Pain Association for their support with this work.

## Author Contributions

**Conceptualization:** Katharina Weßollek, Michael Czaplik, Pascal Kowark.

**Data curation:** Katharina Weßollek.

**Formal analysis:** Katharina Weßollek, Pascal Kowark.

**Investigation:** Katharina Weßollek.

**Methodology:** Katharina Weßollek, Pascal Kowark.

**Project administration:** Ana Kowark, Rolf Rossaint, Pascal Kowark.

**Supervision:** Ana Kowark, Michael Czaplik, Rolf Rossaint, Pascal Kowark.

**Visualization:** Katharina Weßollek, Michael Czaplik.

**Writing – original draft:** Katharina Weßollek.

**Writing – review & editing:** Katharina Weßollek, Ana Kowark, Michael Czaplik, Rolf Rossaint, Pascal Kowark.

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
