## [Decision Letter · Decision Letter 0]

1 Feb 2021

PONE-D-20-35630

The pain drawing as a screening tool for anxiety, depression and reduced health-related quality of life in chronic back pain patients: a cohort study

PLOS ONE

Dear Dr. Kowark,

Thank you for submitting your manuscript to PLOS ONE. After careful consideration, we feel that it has merit but does not fully meet PLOS ONE’s publication criteria as it currently stands. Therefore, we invite you to submit a revised version of the manuscript that addresses the points raised during the review process.

We look forward to receiving your revised manuscript.

Kind regards,

Stephan Doering, M.D.

Academic Editor

PLOS ONE

2. Thank you for including your ethics statement: "The ethics committee of the University Hospital RWTH Aachen, Germany has approved a waiver of consent for this retrospective study (EK 187/18) in July 2018."   

3. In the ethics statement in the manuscript and in the online submission form, please provide additional information about the patient records/samples used in your retrospective study, including: a) whether all data were fully anonymized before you accessed them; b) the date range (month and year) during which patients' medical records/samples were accessed.

Reviewers' comments:

Reviewer's Responses to Questions

**Comments to the Author**

1. Is the manuscript technically sound, and do the data support the conclusions?

Reviewer #1: No

Reviewer #2: Partly

2. Has the statistical analysis been performed appropriately and rigorously? 

Reviewer #1: Yes

Reviewer #2: Yes

3. Have the authors made all data underlying the findings in their manuscript fully available?

Reviewer #1: No

Reviewer #2: Yes

4. Is the manuscript presented in an intelligible fashion and written in standard English?

Reviewer #1: No

Reviewer #2: Yes

5. Review Comments to the Author

Reviewer #1: In general, the paper needs extensive revision of English. Just in the first sentence of the abstract several typos can be observed. In fact, the paper needs to be completely rewritten focusing on low back pain, and not chronic pain.

The abstract needs to be completely rewritten, it is impossible to understand

Introduction: this section needs to be completely rewritten focusing on low back pain, and not chronic pain in general. Specific data on LBP should be included, not about chronic pain in general. In addition, it should be better reorganized.

Methods: No data on inclusion and exclusion criteria are provided. Psychometric data of the outcomes is needed. Definition of cut-off values of HADS scores is needed. Better explanation of how pain drawings were assesses should be also clarified.

Results: It is poorly written. This section should be more scientifically written.

Reviewer #2: The present manuscript reports on the validation of pain drawing, as a screening tool for additional factors like anxiety, depression and quality of life, which are often impaired in pain patients and comorbidity with depression and anxiety is also often high in those patient samples.

The paper is quite well written, the methods are sound, but the authors need to be more specific throughout the manuscript and also need to consider specific variables like age, sex and pain duration in their analyses and interpretation. Otherwise, in my opinion, it is not really a validation study and does then also not really inform the clinical field.

Abstract:

- Subheadings "Results:" and "Conclusions" are missing.

- Would also add information of the pain drawing in the Methods section and also information on the performed analyses.

- Would suggest to specify the direction of the correlations.

- The statement of the authors is not clear for the sentence "Further, we detected.....".

- The authors speak about “calculated values”, but exactly which calculated values / how calculated, and based on which criteria have the authors decided that these "calculated" values are not high enough

- The language in the Abstract, specifically the Results and Conclusions part is a bit vague - since the aim of the study is to determine validity of an instrument for specific factors, the authors need to be clear and specific.

Introduction:

- The authors speak a lot about treatment / the treatment situation in the beginning of the Intro. The patient drawing is tool for diagnostics - this can of course also improve treatment if used for example for treatment success in specific areas or for the evaluation of sub-groups that might need specific treatment, but the background needs to be more related to the characteristics of the pain drawing first and the describe exactly why, i.e. in which way, it is important also for treatment and then why it is needed to know whether it is also associated with pain-related factors like anxiety, depression and quality of life. Here, I would think the authors could speak for example specifically about individual sub-groups for treatment etc.

- The structure and content of the Intro is thus not totally convincing as to why the present study is of such important and has major add on knowledge, particularly also for the treatment of pain patients. Would suggest to re-structure/re-fine the Intro accordingly.

Materials and Methods:

- What about covariates like age, sex and pain duration? These are important aspects that needs to be considered in such a validation study. Also, when thinking about sub-groups for treatment, those variables are also important as variables of interest, and not only as control variables.

Results:

- Patient characteristics would specifically be mean scores with sex and age information – would therefore suggest to add sex and age to the Table and also add respective Figures (separately for male and female and plotted against age for example) for the Supplement

- Would also suggest to include some Results Figures in the main text instead of only Tables to make the results directly visible, including the range of the scores etc.

Discussion and Conclusions:

- As mentioned for the Abstract and Introduction, the authors need to be more specific here in terms of the meaning of such a validation study. Age, sex and pain duration also needs to be discussed here.

6. PLOS authors have the option to publish the peer review history of their article (what does this mean?). If published, this will include your full peer review and any attached files.

Reviewer #1: No

Reviewer #2: No

---

## [Author Response · Author response to Decision Letter 0]

25 May 2021

All reviewer and editor comments are discussed in detail in the file named "Response to Reviewers".

---

## [Decision Letter · Decision Letter 1]

16 Jun 2021

PONE-D-20-35630R1

Pain drawing as a screening tool for anxiety, depression and reduced health-related quality of life in back pain patients: a cohort study

PLOS ONE

Dear Dr. Kowark,

Thank you for submitting your manuscript to PLOS ONE. After careful consideration, we feel that it has merit but does not fully meet PLOS ONE’s publication criteria as it currently stands. Therefore, we invite you to submit a revised version of the manuscript that addresses the points raised during the review process.

We look forward to receiving your revised manuscript.

Kind regards,

Stephan Doering, M.D.

Academic Editor

PLOS ONE

Journal Requirements:

Reviewers' comments:

Reviewer's Responses to Questions

**Comments to the Author**

1. If the authors have adequately addressed your comments raised in a previous round of review and you feel that this manuscript is now acceptable for publication, you may indicate that here to bypass the “Comments to the Author” section, enter your conflict of interest statement in the “Confidential to Editor” section, and submit your "Accept" recommendation.

Reviewer #1: (No Response)

Reviewer #2: All comments have been addressed

2. Is the manuscript technically sound, and do the data support the conclusions?

Reviewer #1: No

Reviewer #2: Yes

3. Has the statistical analysis been performed appropriately and rigorously? 

Reviewer #1: Yes

Reviewer #2: Yes

4. Have the authors made all data underlying the findings in their manuscript fully available?

Reviewer #1: Yes

Reviewer #2: Yes

5. Is the manuscript presented in an intelligible fashion and written in standard English?

Reviewer #1: No

Reviewer #2: Yes

6. Review Comments to the Author

Reviewer #1: The paper ios much improved but still extensive work. The introduction does not jusfity the current study. Muc more data on pain draws and psychological variables in LBP is clerarly needed. An hypothesis is also needed.

The use of restrospective data should be mentioned as a huge limitation. Recognition bias should be discussed.

The results section should be completely rewritten it is highly difficult to read. Authors repeat once and twice the same sentences with different outcomes

The discussion needs extensive revision, subheadings are not clinically.

Reviewer #2: The authors have done a good job in revising the mansucripts - all my comments have been addressed, no further issues left.

7. PLOS authors have the option to publish the peer review history of their article (what does this mean?). If published, this will include your full peer review and any attached files.

Reviewer #1: No

Reviewer #2: No

---

## [Author Response · Author response to Decision Letter 1]

20 Sep 2021

Rebuttal letter PONE-D-20-35630R1

Dear Stephan Doering, Dear Reviewers, 

Thank you for the important comments on our revised manuscript (“Pain drawing as a screening tool for anxiety, depression and reduced health-related quality of life in back pain patients: a cohort study“).

In the following, you will find our point-by-point responses to your comments in green font.

Thank you,

Sincerely, 

Ana Kowark

Journal Requirements:

Response: Thank you, we carefully reviewed our entire reference list and did not find any retracted papers. Reference number 3 (Shaballout N, Aloumar A, Neubert T-A, Dusch M, Beissner F. Digital Pain Drawings Can Improve Doctors’ Understanding of Acute Pain Patients: Survey and Pain Drawing Analysis. JMIR Mhealth Uhealth. 2019;7(1):e11412. doi: 10.2196/11412. Erratum in: JMIR Mhealth Uhealth. 2019 Sep 27;7(9):e16017. PMID: 30632970; PMCID: PMC6329897.) had an erratum. We updated the reference list to include the erratum.

Reviewers' comments:

Reviewer's Responses to Questions

Comments to the Author

1. If the authors have adequately addressed your comments raised in a previous round of review and you feel that this manuscript is now acceptable for publication, you may indicate that here to bypass the “Comments to the Author” section, enter your conflict of interest statement in the “Confidential to Editor” section, and submit your "Accept" recommendation.

Reviewer #1: (No Response)

Reviewer #2: All comments have been addressed

2. Is the manuscript technically sound, and do the data support the conclusions?

Reviewer #1: No

Reviewer #2: Yes

3. Has the statistical analysis been performed appropriately and rigorously?

Reviewer #1: Yes

Reviewer #2: Yes

4. Have the authors made all data underlying the findings in their manuscript fully available?

Reviewer #1: Yes

Reviewer #2: Yes

5. Is the manuscript presented in an intelligible fashion and written in standard English?

Reviewer #1: No

Reviewer #2: Yes

6. Review Comments to the Author

Reviewer #1: The paper ios much improved but still extensive work. The introduction does not jusfity the current study. Muc more data on pain draws and psychological variables in LBP is clerarly needed. An hypothesis is also needed.

Response: Thank you for your remark. We rephrased the last paragraph of the introduction to clarify the hypothesis and to illustrate the gain of knowledge more clearly (“In this study new pain drawing variables, namely, "hatching degree" and "total word count (TWC)" were added to previously investigated pain drawing variables. We hypothesised that a combination of pain drawing variables can be used as a screening tool for signs of reduced health-related quality of life, depression or anxiety in patients with lower back pain.”, page 4, line 91 et seq.). 

We added findings from pertinent systematic reviews to present the current state of research more accurately and clarified that the majority of the cited papers refer to lower back pain patients (“Some studies on patients with lower back and musculoskeletal pain have reported a statistically significant correlation between pain drawing variables, such as pain extent and pain site, and psychological characteristics, such as depression and somatisation [14-16]. Contrary, other studies mainly focussing on lower back pain patients have not found significant results [17-19]. A systematic review and meta-analysis published in 2015 concluded that the use of qualitatively analysed pain drawings as a screening tool for psychiatric conditions in lower back pain patients currently cannot be recommended without restriction [20]. Nevertheless, the authors acknowledged that future meta-analyses might come to a different result by using higher quality and further studies [20].”, page 4, line 81 et. seq.).

The use of restrospective data should be mentioned as a huge limitation. 

Response: Thank you, we have adapted out limitation section as following: “The main limitation of this study is the use of retrospective data.” (page 20, line 413). 

Recognition bias should be discussed.

Response: Thank you. We have added a discussion in the limitations section on information bias and detection bias: (“Retrospective data analyses are scientifically weaker than other study designs such as randomised-controlled trials and pose the risk of information bias. Further, this study was not blinded, resulting in a high risk of detection bias.”, page 20, line 414 et seq.).

The results section should be completely rewritten it is highly difficult to read. Authors repeat once and twice the same sentences with different outcomes

Response: Thank you for this comment. We carefully rewrote the results section to minimise redundancy. We removed findings regarding non-significant results. We shortened the passage on the results of linear regression analyses. The paragraph on the results of logistic regression was rephrased, but the general structure was maintained to ensure comprehensibility. Our manuscript underwent Elsevier language editing previously. 

The discussion needs extensive revision, subheadings are not clinically.

Response: Thank you for your remark. We have used subheadings as recommended in the STROBE statement “Explanation and Elaboration” article (“We recommend that authors structure their discussion sections, perhaps also using suitable subheadings.“) [1]. According to your suggestion we removed the subheadings. 

Additionally, we have restructured the whole discussion and added phrases to make it more cohesive and easier to read. For example, we started with a short summary of the results, as recommended in the STROBE statement “Explanation and Elaboration” article [1]. 

We hope that our edits address your comment sufficiently. Otherwise, we would be grateful for some constructive suggestions for the design of the extensive revision.

1. Strengthening the Reporting of Observational Studies in Epidemiology (STROBE): Explanation and Elaboration

Vandenbroucke JP, von Elm E, Altman DG, Gøtzsche PC, Mulrow CD, et al. (2007) Strengthening the Reporting of Observational Studies in Epidemiology (STROBE): Explanation and Elaboration. PLOS Medicine 4(10): e297. https://doi.org/10.1371/journal.pmed.0040297

Reviewer #2: The authors have done a good job in revising the mansucripts - all my comments have been addressed, no further issues left.

7. PLOS authors have the option to publish the peer review history of their article (what does this mean?). If published, this will include your full peer review and any attached files.

Do you want your identity to be public for this peer review? For information about this choice, including consent withdrawal, please see our Privacy Policy.

Reviewer #1: No

Reviewer #2: No

While revising your submission, please upload your figure files to the Preflight Analysis and Conversion Engine (PACE) digital diagnostic tool, https://pacev2.apexcovantage.com/. PACE helps ensure that figures meet PLOS requirements. To use PACE, you must first register as a user. Registration is free. Then, login and navigate to the UPLOAD tab, where you will find detailed instructions on how to use the tool. If you encounter any issues or have any questions when using PACE, please email PLOS at figures@plos.org. Please note that Supporting Information files do not need this step

---

## [Editor Report · Decision Letter 2]

27 Sep 2021

Pain drawing as a screening tool for anxiety, depression and reduced health-related quality of life in back pain patients: a cohort study

PONE-D-20-35630R2

Dear Dr. Kowark,

We’re pleased to inform you that your manuscript has been judged scientifically suitable for publication and will be formally accepted for publication once it meets all outstanding technical requirements.

Kind regards,

Stephan Doering, M.D.

Academic Editor

PLOS ONE

---

## [Editor Report · Acceptance letter]

1 Oct 2021

PONE-D-20-35630R2 

Pain drawing as a screening tool for anxiety, depression and reduced health-related quality of life in back pain patients: a cohort study 

Dear Dr. Kowark:

I'm pleased to inform you that your manuscript has been deemed suitable for publication in PLOS ONE. Congratulations! Your manuscript is now with our production department. 

Kind regards, 

on behalf of

Professor Stephan Doering 

Academic Editor

PLOS ONE